# Temporal Variation of Meropenem Resistance in *E. coli* Isolated from Sewage Water in Islamabad, Pakistan

**DOI:** 10.3390/antibiotics11050635

**Published:** 2022-05-09

**Authors:** Saba Yasmin, Asad-Mustafa Karim, Sang-Hee Lee, Rabaab Zahra

**Affiliations:** 1Department of Microbiology, Quaid-i-Azam University, Islamabad 45320, Pakistan; sabauvas123@gmail.com; 2National Leading Research Laboratory of Drug Resistance Proteomics, Department of Biological Sciences, Myongji University, Yongin 17058, Korea; asadmustafa8@gmail.com

**Keywords:** meropenem resistance, *E. coli*, temporal variation, environment, MCR genes, Pakistan

## Abstract

The WHO has classified carbapenem-resistant Enterobacteriaceae in most critical priority pathogens that pose a threat to human health. The present study investigated the prevalence of meropenem-resistant *Escherichia coli* (*E. coli*) in relation to its temporal variation in different seasons along with its resistance markers in sewage water. *E. coli* was selected on MacConkey agar containing meropenem (3 µg/mL). There were 27% of sites/sewage samples carrying meropenem-resistant *E. coli*. All *E. coli* were confirmed through the amplification of the *uid*A gene. All isolated *E. coli* were multidrug-resistant (MDR), and among them, 51% were extensively drug-resistant (XDR). An antibiogram determined against 15 antibiotics showed the highest resistance to ampicillin and cefotaxime (98% each) and lowest resistance to fosfomycin (2%). Phylogenetic groups and resistance gene analysis through PCR showed a significant co-occurrence of carbapenemases with extended spectrum beta lactamases (ESBLs), plasmid encoded quinolone, and colistin resistance genes. The higher number of resistance genes in *E. coli* isolates in community sewage indirectly indicate that these isolates circulate abundantly in the community.

## 1. Introduction

Bacterial infectious diseases are a leading cause of morbidity and mortality worldwide. The carbapenem-resistant Enterobacteriaceae has been listed among priority 1 pathogens and is a critical threat to human health as declared by the World Health Organization (WHO) [1]. Carbapenems are considered last resort antibiotics, when quinolones and cephalosporins are not effective to treat the infections. The presence of carbapenem resistance in *E. coli* emphasizes the need for their effective surveillance and interventions to prevent their spread in the community [2]. Carbapenem-resistant *E. coli* are considered a critical threat because strains expressing carbapenemases often harbor resistance mechanisms against several other antibiotics, giving them a broad resistance spectrum [3]. Multiple genes contribute towards carbapenem resistance, including those encoding extended spectrum beta lactamases (ESBLs) and carbapenemases, e.g., *bla*_CTX-M_, *bla*_OXA-like_, *bla*_SHV_, *bla*_TEM_, *bla*_VIM_, *bla*_IMP_, *bla*_NDM_, and *bla*_KPC_ [4], which can be on a mobile genetic element and able to horizontally transfer to other species in the environment [5].

Surveillance is an important tool used to manage antibiotic resistance. Up to date information on local resistance patterns is crucial for indicating the usefulness and efficacy of antibiotics [6]. For surveillance, sewage water is considered to be a reliable indicator of community carriage as it contains fecal bacteria from individuals in the community. It opens the possibilities to generate antibiotic resistance data at the community level without the need to sample individuals. It also avoids ethical challenges associated with individual screenings, such as stigmatization of carriers of resistant bacteria [7]. Antibiotic-resistant bacteria are abundant in environmental water systems because the water cycle continually picks and disseminates the microbes through the ecosystem. Sewage water contains antibiotics, biocides, heavy metals, and other chemicals naturally present in it to provide a selection pressure for the resistant bacteria. Wastewater can also act as a mixing vessel and reservoir for transferring antimicrobial resistance genes (ARGs) to co-residing micro-organisms, such as *E. coli* [8]. Numerous environmental factors including biotic (e.g., microbial communities and bacteriophages) and abiotic factors (temperature, water, nutrient availability, pH, and solar radiation) contribute to the survival and spread of resistant bacteria in the environment [9]. Factors involved in the spread of resistant bacteria are likely to change with seasonal changes, rainfall variability, and human activities which may contaminate surface and drinking waters [10]. Considering the potential of horizontal ARG transfer to reservoirs of human pathogens, the identification/surveillance of ARGs in bacteria isolated from environmental waters is gaining importance [11]. Monitoring of antimicrobial resistant bacteria in wastewater could be used to predict clinical antibiotic resistance levels that in turn could provide precise directions for antibiotic use and antimicrobial resistance (AMR) management [12].

Keeping in view the clinical importance of carbapenem-resistant *E. coli*, and the role of sewage water in the prevalence of antibiotic-resistant bacteria, the present study was designed to monitor sewage water for the presence of antibiotic-resistant *E. coli*, to understand the phenotypic and genotypic resistance profiles of *E. coli* isolates, and to examine their phylogenetic groups with seasonal variations. *E. coli*, being a fecal indicator due to its extensive presence in wastewater as excreted from the gut of humans and animals, have been used for microbial source tracking; since antibiotics are conventionally used to treat *E. coli* infections in people and animals, this can provide an understanding of AMR status in the environment [13,14]. Thus, by looking into antibiotic-resistant *E. coli* in the environment, we can obtain insights into community spread of antibiotic-resistant bacteria and potential hazards for the community.

## 2. Results

A total of 240 (60 samples/season) sewage samples were collected over four seasons from October 2016 to September 2017. Of these, 27% (*n* = 64) were positive for meropenem-resistant *E. coli*. From 64 sewage samples, there was a total of 124 confirmed meropenem-resistant *E. coli* isolates. The prevalence of meropenem-resistant *E. coli* across the same 60 sites varied in different seasons. The highest number of sites showing meropenem-resistant *E. coli* was in summer at 50% (*n* = 30, *E. coli* isolates = 50) and lowest in winter at 17% (*n* = 10, *E. coli* isolates = 20). Meanwhile, prevalence in fall was 18% (*n* = 11, *E. coli* isolates = 30) and in spring was 22% (*n* = 13, *E. coli* isolates = 24), as shown in Figure 1. Average temperatures during sampling were as follows: fall, 16–21 °C; winter, 8–12 °C; spring, 16–23 °C; and summer, 25–31 °C. There was not a single site that continually carried meropenem-resistant *E. coli* in all four seasons. Details of each site along with number of meropenem-resistant *E. coli* isolates are shown in Table 1.

### 2.1. Antibiotic Susceptibility Profiles

All study isolates were subjected to a disc diffusion test to determine their susceptibility against 15 different antibiotics belonging to 13 different antibiotic classes. All isolates were found to be MDR (resistant to ≥3 tested antibiotic classes). Among all MDR isolates, 51% (*n* = 63) were XDR (susceptible to ≤3 tested antibiotic classes), while there was no PDR isolate (resistant to all 15 tested antibiotics). Comparatively higher percentages of XDR were found in spring (79%) and summer (56%) compared to winter (20%) and fall (40%). The antibiotic resistance profile of study isolates against all tested antibiotics is as follows: ampicillin, 98%; ceftazidime, 86%; ampicillin–clavulanic acid, 94%; cefotaxime, 98%; cefepime, 84%; aztreonam, 85%; meropenem, 71%; ciprofloxacin, 93%; chloramphenicol, 35%; nalidixic acid, 91%; gentamicin, 54%; doxycycline, 62%; nitrofurantoin, 15%; fosfomycin, 2%; and sulfamethoxazole-trimethoprim, 92%. We observed intermediate resistance to doxycycline (11%), meropenem (29%), and fosfomycin (29%).

The temporal variation of antibiotic susceptibility profile of study isolates is shown in Figure 2. *E. coli* isolated in summer showed a slightly different pattern in comparison to other season’s *E. coli* isolates. Only summer *E. coli* showed resistance to fosfomycin (4%, *n* = 2). *E. coli* isolates of fall and winter were 100% resistant to meropenem as well. However, in spring and summer, 79% and 38% isolates were resistant and 21% and 62% were intermediately resistant, respectively.

### 2.2. Phylogenetic Groups of E. coli

Among the typeable isolates, the following phylogenetics groups were found: B1 (2%), B2 (6%), Clad I/II (38%), E/cladI/II (2%), A/C (7%), and F (~1%). *E. coli* isolated in the summer season showed a higher diversity of phylogenetic groups, and F phylogroup was only observed in summer. On the other hand, B1 isolates were only found in fall. There were 44% isolates un-typeable by quadruplex PCR and remain unknown. Seasonal prevalence comparison of *E. coli* phylogenetic groups is shown in Table 2.

### 2.3. Molecular Screening of Antimicrobial Resistance Markers

All *E. coli* were screened to obtain an insight into their resistance markers encoding carbapenemases, ESBLs, plasmid-mediated quinolones resistance (PMQRs), and plasmid-mediated colistin-resistance genes. Among carbapenemases, the most prevalent β-lactamases were *bla*_OXA-48_ (65%) and *bla*_IMP_ (34%), followed by *bla*_NDM_ (25%), *bla*_KPC_ (~1%), and *bla*_BIC_ (~1%). Among ESBLs, the most prevalent was *bla*_TEM_ (55%), followed by *bla*_CTX-M_ (46%), and *bla*_SHV_ (15%). Among PMQR markers, the most prevalent was *qnr*S (32%), followed by *qnr*B (17%) and *qnr*A (3%). Among the *mcr* group of genes (from *mcr*-1 to *mcr*-5), only *mcr*-1 (35%) was found among study isolates.

The comparative prevalence of resistance markers in four seasons is shown in Table 2. Summer showed co-occurrence of 12 resistance markers, and *bla*_KPC_ and *bla*_BIC_ were only found in summer. Among the 16 screened ARGs, the lowest co-occurrence was found among winter isolates. None of the isolates in winter were positive for *qnr*A, *qnr*B, *bla*_KPC,_ or *bla*_BIC_, but the highest prevalence of *mcr*-1 was in winter.

There were 5% (*n* = 6) isolates that carried all four classes of resistance genes (ARGs), with at least one of the carbapenemases, ESBL, PMQRs, and *mcr* group of genes. Moreover, there were 5% (*n* = 6) isolates which did not show the presence of any of the screened ARGs by PCR. All isolates were MDR phenotypically, and there were 44% (*n* = 55) *E. coli* carrying a combination of three classes of ARGs. The highest number of study isolates were carrying a combination of carbapenemases, ESBLs, and PMQRs. Isolates categorized based on the combination of carriage of antibiotic resistance genes are shown in Figure 3.

## 3. Discussion

The present study was carried out in order to obtain insights into the prevalence and temporal variation of meropenem-resistant *E. coli* in sewage water, along with their carriage of resistance genes. The study showed that 27% of the sites/sewage samples were positive for meropenem-resistant *E. coli*, and a comparatively higher prevalence was observed in summer. In contrast, a previous study from Islamabad of *E. coli* isolated from wastewater reported 100% carbapenem susceptibility of *E. coli* isolated from sewage water [15]. Another study from the neighboring country India reported only 3% carbapenem-resistant *E. coli* from sewage water [16]. The difference between studies suggests an increasing resistant *E. coli* burden in the environment over time. An increasing AMR trend in the environment can be correlated with increasing antibiotic usage over the time period. A global study published in 2018 reported comprehensive data of antibiotic usage in 76 countries over a period of 16 years (2000–2015). Overall, there was a 65% increase in antibiotic consumption. Among low- and middle-income countries, it was found that Pakistan (65% growth) was the 3rd largest among those with an increasing trend of antibiotic usage after India (103% growth) and China (79% growth). Of particular concern was the increase in usage of last-resort antibiotic classes, which was same in low-, middle-, and high-income countries, including carbapenems, polymyxins, glycylcyclins, and oxazolidinones. In particular, usage of carbapenems and polymyxins (largely colistin) increased during the study period in low- and middle-income countries. [17]. Another possible reason could be due to antibiotic-based selection of *E. coli* in the present study, which better enabled us to sort and isolate meropenem-resistant *E. coli*.

The present study reports 51% XDR. The most effective drugs were fosfomycin, nitrofurantoin, and chloramphenicol. Resistance to fluoroquinolones, quinolones, and cephalosporins was above 80%. One study from India found a maximum of 60% *E. coli* isolates to be resistant to fluoroquinolones, with less than 60% to quinolones and cephalosporins, and the most effective drug was carbapenem [16]. The difference of phenotypic antibiotic resistance in the present study in comparison to previous reports could be because the present study is focused on meropenem-resistant isolates, which are known for the co-occurrence of resistance markers against other antibiotics [5]. The comparatively lower resistance to chloramphenicol (35%) could be due to its lower level of use/prescription in comparison to other antibiotics. The present study reports higher susceptibility to fosfomycin. This could be because fosfomycin is among the less often prescribed antibiotics and is reserved as the last treatment option for infections caused by carbapenemase-producing Enterobacteriaceae [18].

In phylogenetic analysis, 44% of *E. coli* were un-typeable, and the typeable isolates mostly belonged to clad I/II, which suggests that most of the *E. coli* were environmental/commensals. A study [16] from India reported that most of the antibiotic-resistant *E. coli* belonged to the B2 phylogenetic group (67%), which is considered a virulent/pathogenic group. In contrast, the present study reported only 6% of MDR *E. coli* belonging to the B2 group. This contradiction shows that resistant isolates do not necessarily belong to the B2 group, but can be commensals. This difference could be because the previous study was only based on hospital wastewater, which can carry more pathogenic *E. coli* than other sites. These results also draw attention towards the level of horizontal gene transfer of antibiotic resistance genes from pathogenic to commensals in the environment, making them MDR and XDR.

Results of ARGs showed the co-prevalence of multiple resistance genes with carbapenemases. *bla*_CTX-M_-type ESBLs have been reported worldwide as the primary cause of resistance to third-generation cephalosporins in *E. coli* isolated from clinical samples and commensals [19]. The human colonization of ESBL-producing *E. coli* is estimated to be around 14% globally, with rates as high as 22% in Asia and Sub-Saharan Africa [20]. The present study reported 15–55% ESBLs from environmental *E. coli*, which showed a comparatively higher ARG burden in the environment than previous studies. A recent *E. coli* study from wastewater reported 84% *bla*_TEM_, followed by 52% *bla*_CTX-M_ and no *bla*_SHV_. In the present study, a similar pattern was found, where the most prevalent ESBL gene was *bla*_TEM_ (55%), followed by *bla*_CTX-M_ (46%). Among carbapenemases, the other study found 16% each for *bla*_NDM-1_ and *bla*_KPC-2_ [21]. In the present study, carbapenemase prevalence is quite different, as the highest level of carbapenemase found was *bla*_OXA-48_ (65%), followed by *bla*_IMP_ (34%), *bla*_NDM_ (25%), and ~1% *bla*_KPC_. The difference in carbapenemase pattern among *E. coli* in different geographical regions could be due to different prevalent sequence types (STs) with different resistance mechanisms.

Previous studies from Pakistan reported *mcr*-1 in *E. coli* isolated from wild migratory birds [22], humans [23], and broiler chickens [24] over the last 5 years. Recently, a study form Pakistan reported less than 1% *mcr*-1 co-carriage with ESBLs in clinical isolates [25], while we found 9% of *E. coli* co-harboring *mcr*-1 with carbapenemases, ESBLs, and PMQRs. Theearlier preliminary reports from Pakistan were during 2016–2018, while the present study conducted on sewage water samples, collected during 2016–2017, found 35% *mcr*-1 in *E. coli*, indicating that there was much more colistin resistance burden in the environment than that found in clinical or other settings, which is a point of deep concern.

*E. coli* isolated in summer showed maximum diversity of phylogenetic groups and carriage of antimicrobial resistance genes. The comparatively greater prevalence of meropenem-resistant *E. coli* in summer could be due to favorable and survival-supporting environmental factors, including temperature; in summer, the mean environmental temperature is above 30 °C in plain areas of Pakistan, which is optimum for bacterial growth. Another possible reason could be the higher infection rate of meropenem-resistant *E. coli* in the community in summer, which ultimately disseminates in sewage water through fecal contamination. A study from India investigating pathogen inhabitant rate in water in different seasons found the presence of total and thermotolerant coliforms in different microenvironments of water, which were highest in the monsoon, followed by summer, and least in winter [26]. One retrospective study from Pakistan reported more bacterial pathogens in stool samples collected from patients in summer. They reported Vibrio cholera 01 Ogawa (32.8%), *Campylobacter jejuni* (17.3%), Enteropathogenic *E. coli* (9.9%), Salmonella paratyphi b (6.6%), and *Shigella flexneri* (6.2%). The study suggested seasonal variation in organism detection, demonstrating an inclination towards summer [27]. Diverse types of phylogenetic groups and ARGs in summer could be because of the higher number of *E. coli* isolated in summer, increasing the probability of variation. 

The findings of the present study, including detection of high phenotypic antibiotic resistance patterns and ARGs in environmental commensal *E. coli*, are thought provoking. These commensals are expected to be carried in the gut of humans and animals and disseminate into the environment through fecal contamination, where these MDR commensal *E. coli* can easily transfer plasmid-mediated ARGs to other co-residing bacteria in environment. Through sewage water, antibiotic resistance can disseminate and reach all living beings (human, animals, and birds) and non-living things (soil, agriculture, and the rest of the water system) which can also ultimately become disseminators. It can become part of an ecosystem and disseminate more extensively than the control measures can counter. Thus, continuous checks of the water to gain an insight into AMR and taking prompt actions for its specified treatment to break the chain of dissemination are most necessary.

## 4. Materials and Methods

### 4.1. Sampling

A total of 60 different sites were targeted across the capital city of Islamabad, Pakistan, as shown in the map (Figure 4). The sites included educational institutes, hospitals, commercial markets, office buildings, residential areas, and downstream waterfalls (Nullahs). From each site highlighted, 30–45 mL of sewage water was collected during October 2016–September 2017, covering all four seasons (fall, winter, spring, and summer).

### 4.2. Isolation and Identification of Meropenem-Resistant E. coli

Sewage samples were spun down at 1500 rpm for 3–5 min to remove visible garbage. From each sample, 100 µL water was added to screening agar (MacConkey agar supplemented with 3 µg/mL meropenem) by the spread plate method and incubated at 37 °C for 24 hrs. Suspected *E. coli* colonies were processed for confirmation by standard biochemical tests and through amplification of *uid*A gene by using primers as mentioned in Table 1. All confirmed *E. coli* isolates were preserved in 20% glycerol at −80 °C for further study. 

### 4.3. Antibiotic Susceptibility Test

All study isolates’ susceptibility profiles were analyzed against 15 different antibiotics using the Kirby–Bauer disk diffusion method [28]. Zone diameters were interpreted according to CLSI 2019 guidelines. Antibiotic discs and their concentrations were as follows: Ampicillin (AMP 10 µg), ceftazidime (CAZ 30 µg), amoxicillin–clavulanic acid (AMC 10/20 µg), cefotaxime (CTX 30 µg), cefepime (FEP 30 µg), aztreonam (ATM 30 µg), meropenem (MEM 10 µg), ciprofloxacin (CIP 5 µg), Chloramphenicol (C 30 µg), Nalidixic acid (NA 30 µg), Gentamicin (CN 120 µg), Doxycycline (DO 30 µg), Nitrofurantoin (F 300 µg), Fosfomycin (FOT 200 µg), and trimethoprim-sulfamethoxazole (SXT 1.25/25 µg).

### 4.4. DNA Extraction

Whole genomic extraction of *E. coli* was performed using the boil colony method. The overnight 3–5 colonies of *E. coli* were suspended in 200 μL sterile distilled water. The suspension was thoroughly mixed by vortex, boiled at 95 °C for 15 min, and subsequently centrifuged at 14,000 rpm for 8 min. The supernatant was collected and stored at −20 °C for further processing [29].

### 4.5. Phylogenetic Analysis of E. coli

All study isolates were subjected to quadruplex PCR for phylogenetic analysis using the Clermont method [30].

### 4.6. Molecular Screening for Antibiotic Resistance Genes

All isolates were screened for ARGs using conventional singleplex and multiplex PCRs. Primers used along with their annealing temperature are given in Table 3. The PCR cycle conditions were as follows.

The reaction mixture contained 12.5 μL of 2X mastermix (Thermo scientific Dream *Taq*^Tm^ green PCR master mix), 0.2 μL each of forward and reverse primers (100 pmoles/µL), 2 μL of DNA template, and PCR-grade water up to the final volume of 25 μL. The cycling conditions for the PCR reaction were as follows: an initial denaturation for 10 min at 94 °C, followed by 35 cycles containing denaturation at 94 °C for 40 sec, annealing for 40 sec, initial extension at 72 °C for 1 min, and a final extension for 5 min at 72 °C. Agarose gel electrophoresis was performed at 90 V for 55 min in 1.5% agarose gel made in 10X Tris boric acid-EDTA (TBE) buffer.

## 5. Conclusions

The present study was focused on surveillance of meropenem-resistant *E. coli* along with other resistance mechanisms in wastewater. From our findings, we suggest conducting integrating research to elucidate their genetic details, sequence types, and study of environmental factors contributing to their successful survival in environmental water. Wastewater carriage of MDR *E. coli* along with combinations of multiple resistance genes (including carbapenemases and plasmid-mediated colistin resistance) is an early warning sign of potential hazard for the community, so the present study concerns public health, calling for a comprehensive monitoring and control system for clinically important antibiotic-resistant bacteria in the environment. Through control measures, antibiotic-resistant bacteria circulating in the environment should be minimized to prevent community-acquired infections, occupational hazards, and a possible public health hazard.

## Figures and Tables

**Figure 1 antibiotics-11-00635-f001:**
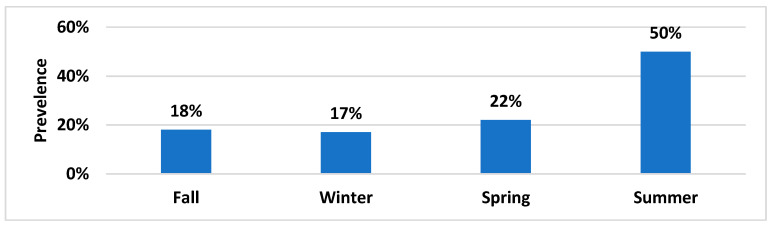
Temporal prevalence of meropenem-resistant *E. coli* in sewage samples.

**Figure 2 antibiotics-11-00635-f002:**
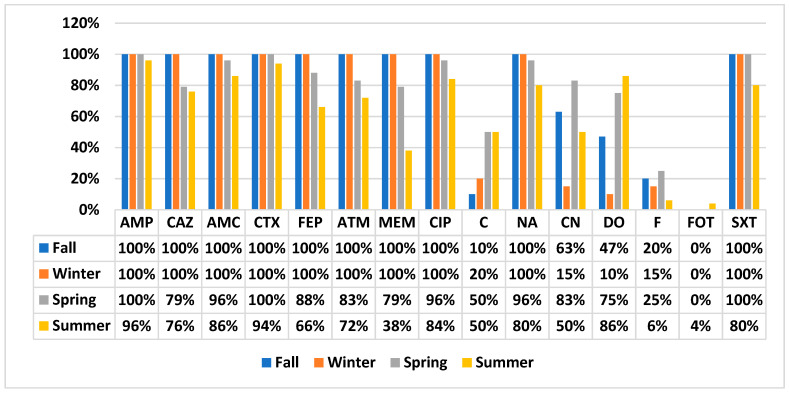
Temporal variation in antibiotic susceptibility profile of *E. coli*.

**Figure 3 antibiotics-11-00635-f003:**
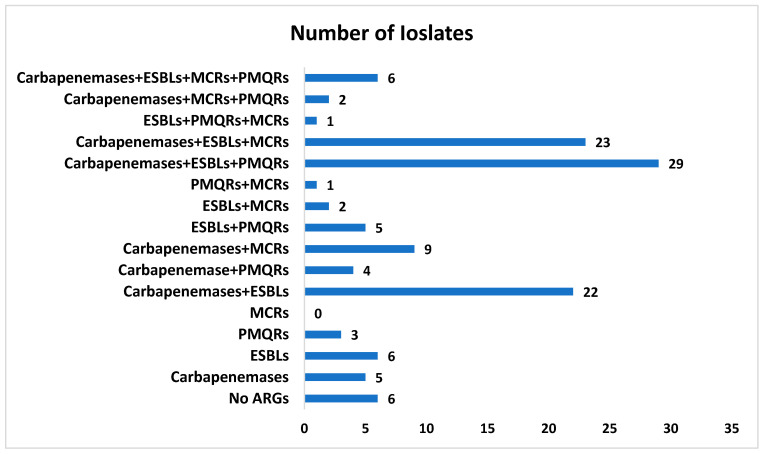
The number of isolates carrying a combination of antibiotic resistance genes.

**Figure 4 antibiotics-11-00635-f004:**
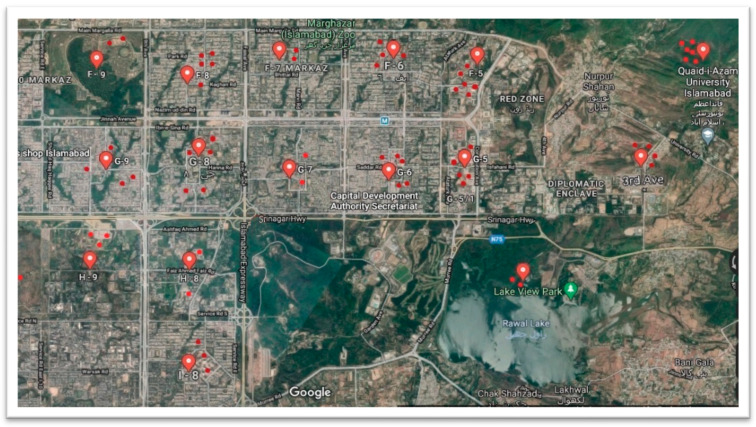
Map showing sites of sewage water samples across the capital city Islamabad.

**Table 1 antibiotics-11-00635-t001:** Meropenem-resistant *E. coli* in different seasons of each site/sample.

Sewage Sampling in Different Seasons				
Site/Location	Fall	Winter	Spring	Summer
C-type colony, QAU	–	–	–	+ (*n* = 1)
B-type colony, QAU	–	–	–	+ (*n* = 2)
QAU D-type colony entry point of filtration plant	–	–	–	+ (*n* = 6)
QAU D-type colony after filtration plant	–	–	–	–
Bari imam Nullah 1	+ (*n* = 1)	–	–	+ (*n* = 1)
Bari imam Nullah 2	–	–	–	+ (*n* = 4)
Sector G-5 Nullah	–	–	–	+ (*n* = 2)
Sector G-5 Fata house exit	+ (*n* = 1)	–	–	–
Ministry of water and power indoor	+ (*n* = 1)	+ (*n* = 1)	–	–
Govt. hostel lodge 2	–	–	–	+ (*n* = 2)
Federal lodge 1	–	–	–	+ (*n* = 1)
MNA HOSTEL	–	–	+ (*n* = 1)	+ (*n* = 2)
PM office indoor	–	–	+ (*n* = 1)	
PARC head office indoor	+ (*n* = 1)	+ (*n* = 6)	–	+ (*n* = 1)
Ambassador hotel	–	+ (*n* = 1)	–	–
Ministry of climate change indoor	–	+ (*n* = 1)	+ (*n* = 2)	+ (*n* = 2)
PHRC indoor	–	+ (*n* = 2)	–	+ (*n* = 3)
Sector G-5 Comsats indoor	–	+ (*n* = 2)	+ (*n* = 4)	+ (*n* = 2)
Sector G-6 Nullah 1	–	+ (*n* = 1)		+ (*n* = 1)
Sector G-6 Nullah 2	–	–	–	–
ICB Islamabad indoor	+ (*n* = 6)	–	+ (*n* = 1)	+ (*n* = 2)
CDA indoor	–	+ (*n* = 3)	–	–
Masjid road Nullah	–	–	–	–
Lal masjid indoor	–	–	–	+ (*n* = 1)
G-6 girls Govt. School	–	–	–	–
G6 Residential outdoor	–	–	–	+ (*n* = 1)
Sector F-6 Nullah	–	–	–	–
Sector F-5 Nullah	+ (*n* = 1)	–	–	+ (*n* = 1)
Baluchistan house indoor	–	–	+ (*n* = 1)	–
PTV office indoor	–	–	–	–
FPSC office indoor	–	–	–	+ (*n* = 1)
KPK house indoor	–	–	+ (*n* = 1)	+ (*n* = 1)
Sindh house indoor	+ (*n* = 1)	–	–	+ (*n* = 1)
Punjab house indoor	+ (*n* = 2)	–	–	+ (*n* = 2)
F-6/3 primary school		–	–	–
F-6/3 masjid		–	–	–
Poly clinic outdoor	+ (*n* = 11)	–	–	–
Poly clinic emergency	+ (*n* = 2)	–	–	+ (*n* = 1)
Nullah F-7	–	–	–	–
GPO indoor	–	+ (*n* = 2)	+ (*n* = 1)	–
Sector H8-sector Nullah	+ (*n* = 3)	–	–	+ (*n* = 1)
Preston university indoor	–	–	–	–
Nullah back side of Turk school	–	+ (*n* = 1)	–	–
Pak Turk school indoor	–	–	+ (*n* = 2)	–
Preston university front Nullah	–	–	–	+ (*n* = 2)
Shah-Abdul Latif auditorium indoor	–	–	+ (*n* = 1)	+ (*n* = 2)
HEC indoor	–	–	–	–
Institute of science and Tech. indoor	–	–	+ (*n* = 6)	–
Post graduate hostel	–	–	–	–
Post graduate college outdoor	–	–	+ (*n* = 1)	–
AIOU indoor	–	–	–	–
Red Crescent indoor	–	–	–	+ (*n* = 1)
Fire brigade	–	–	+ (*n* = 2)	
QAU.GH.2	–	–	–	+ (*n* = 1)
QAU.BH 8	–	–	–	–
QAU.BH.3	–	–	–	–
QAU.BH.4	–	–	–	–
QAU.BH6.7	–	–	–	+ (*n* = 1)
QAU.GH.5	–	–	–	–
QAU colony	–	–	–	+ (*n* = 1)

– = No isolates detected; + = sample carrying meropenem-resistant *E. coli*; *n* = number of *E. coli* isolates detected; MNA—Member of National Assembly; PM—Prime Minister; PARC—Pakistan Agricultural Research Council; PHRC—Pakistan Health Research Council; CDA—Capital Development Authority; PTV—Pakistan Television Corporation; FPSC—Federal Public Service Commission; KPK—Khyber Pakhtunkhwa; GPO—Pakistan Post office; HEC—Higher Education Commission; AIOU—Allama Iqbal Open University; QAU—Quaid-i-Azam University; GH–Girls Hostel; BH–Boys Hostel.

**Table 2 antibiotics-11-00635-t002:** Antibiotic resistance-coding genes present in the *Escherichia coli* isolates from sewage water in different seasons in Islamabad, Pakistan, and their phylogenic groups.

	Sewage Water Samples in Different Seasons				
Antibiotic Resistance Genes	Fall, *n* = 93	Winter, *n* = 64	Spring, *n* = 87	Summer, *n* = 164	*p*-Value
	*n* (%)	*n* (%)	*n* (%)	*n* (%)	
*qnr*A	1 (3)	0	1 (4)	2 (4)	0.4
*qnr*B	7 (23)	0	4 (17)	10 (20)	0.6
*qnr*S	5 (17)	3 (15)	11 (46)	21 (42)	0.7
*bla* _OXA48_	20 (66)	17 (85)	10 (42)	34 (68)	0.67
*bla* _TEM_	15 (50)	13 (65)	18 (75)	22 (44)	0.56
*bla* _SHV_	2 (7)	2 (10)	8 (33)	6 (12)	0.25
*Bla* _CTX-M_	4 (13)	12 (60)	14 (58)	27 (54)	0.77
*bla* _IMP_	14 (46)	5 (25)	8 (33)	15 (30)	0.65
*bla* _NDM_	11 (36)	2 (10)	8 (33)	10 (20)	0.48
*bla* _BIC_	0 (0)	0 (0)	0 (0)	1 (2)	0.32
*bla* _KPC_	0 (0)	0 (0)	0 (0)	1 (2)	0.32
*mcr* *-1*	14 (47)	10 (50)	5 (21)	15 (30)	0.55
*mcr* *-2*	0 (0)	0 (0)	0 (0)	0 (0)	-
*mcr* *-3*	0 (0)	0 (0)	0 (0)	0 (0)	-
*mcr* *-4*	0 (0)	0 (0)	0 (0)	0 (0)	-
*mcr* *-5*	0 (0)	0 (0)	0 (0)	0 (0)	-
		Phylogenic groups			
B1	2 (7)	0 (0)	0 (0)	0 (0)	0.32
B2	7 (23)	0 (0)	0 (0)	1 (2)	0.4
Clad I/II	7 (23)	7 (35)	6 (25)	27 (54)	0.7
Unknown	14 (47)	13 (65)	12 (50)	15 (30)	0.5
E/clad II	0 (0)	0 (0)	2 (8)	1 (2)	0.34
A/C	0 (0)	0 (0)	4 (17)	5 (10)	0.33
F	0 (0)	0 (0)	0 (0)	1 (2)	0.3

*p*-values were extracted from Fisher test. *n* = number of *E. coli* isolates in water samples.

**Table 3 antibiotics-11-00635-t003:** Primer sequences and their product size.

	Primer	Primer Sequence 5′ to 3′	Annealing Temperature	Product Size	Reference
1.	*uid*A	F:5′-AAAACGGCAAGAAAAAGCAG-3′	55 °C	147 bp	[31]
		R:5′-ACGCGTGGTTACAGTCTTGCG-3′			
2.	*bla* _NDM_	F:5′-GGTTTGGCGATCTGGTTTTC-3	52 °C	621 bp	[32]
		R:5′-CGGAATGGCTCATCACGATC-3			
3.	*bla* _IMP_	F: 5′-GAATAGAATGGTTAACTCTC-3′	52 °C	188 bp	[33]
		R:5′-CCAAACCACTAGGTTATC-3′			
4.	*bla* _OXA-48_	F:5′-GCGTGGTTAAGGATGAACAC-3′	52 °C	438 bp	[34]
		R:5′-CATCAAGTTCAACCCAACCG-3′			
5.	*bla* _KPC_	F:5′-CGTCTAGTTCTGCTGTCTTG-3′	52 °C	798 bp	[34]
		R:5′-CTTGTCATCCTTGTTAGGCG-3′			
6.	*bla* _BIC_	F:5′-TATGCAGCTCCTTTAAGGGC-3′	52 °C	537 bp	[34]
		R:5′-TCATTGGCGGTGCCGTACAC-3′			
7.	*bla* _CTX-M_	F:5′-ATGTGCAGTACCAGTAAGGT-3′	52 °C	594 bp	[35]
		R:5′-TGGGTAAAGTAGGTCACCAGA-3′			
8.	*bla* _TEM_	F:5′-CTTCCTGTTTTTGCTCACCCA-3′	52 °C	717 bp	[36]
		R:5′-TACGATACGGGAGGGCTTAC-3′			
9.	*bla* _SHV_	F:5′-TCAGCGAAAAACACCTTG-3′	52 °C	471 bp	[36]
		R:5′-TCCCGCAGATAAATCACC-3′			
10.	*qnr*A	F:5′-AGAGGATTTCTCACGCCAGG-3′	54 °C	580 bp	[37]
		R:5′-TGCCAGGCACAGATCTTGAC-3′			
11.	*qnr*B	F:5′-GGMATHGAAATTCGCCACTG-3′	54 °C	264 bp	[37]
		R:5′-TTTGCYGYYCGCCAGTCGAA-3′			
12.	*qnr*S	F:5′-GCAAGTTCATTGAACAGGGT-3′	54 °C	428 bp	[37]
		R:5′-TCTAAACCGTCGAGTTCGGCG-3′			
13.	*mcr*-1	F:5′-AGTCCGTTTGTTCTTGTGGC-3′	55 °C	320 bp	[38]
		R:5′-AGATCCTTGGTCTCGGCTTG-3′			
14.	*mcr*-2	F:5′-CAAGTGTGTTGGTCGCAGTT-3′	55 °C	715 bp	[38]
		R:5′-TCTAGCCCGACAAGCATACC-3′			
15.	*mcr*-3	F:5′-AAATAAAAATTGTTCCCCGCTTATG-3′	55 °C	929 bp	[38]
		R:5′-AATGGAGATCCCCGTTTTT-3′			
16.	*mcr*-4	F:5′-TCACTTTCATCACTGCGTTG-3′	55 °C	1116 bp	[38]
		R:5′-TTGGTCCATGACTACCAATG-3′			
17.	*mcr*-5	F:5′-ATGCGGTTGTCTGCATTTATC-3′	55 °C	1644 bp	[38]
		R:5′-CATTGTGGTTGTCCTTTTCTG-3′			

## Data Availability

Not applicable.

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
