# Peer review of "Temporal Variation of Meropenem Resistance in E. coli Isolated from Sewage Water in Islamabad, Pakistan"

_antibiotics, 2022, doi:10.3390/antibiotics11050635_

Round 1

Reviewer 1 Report

The submitted paper presents new results with a high level of meropenem resistance in E. coli isolated from wastewater samples in Pakistan. The isolated strains have been well characterized.

Some questions are remaining:

Why 124 isolates were obtained from  64 positive samples? Is the distribution homogenous? When 2 isolates are obtained from the same sample, are the resistance patterns identical? If not, more isolates need to be obtained to check homogeneity of each sample.

As the isolation rate is not homogenous following the seasons, expression in percentage seems not the best way. The number of isolates in each group are rather different. Indeed in figure 5, the number of isolates is indicated (percentages are no longer used). Pay attention to the spelling of carba... in this figure, in the first five lines, errors occurred.

Many of the indicated references are not recent, are further recent data available?

In the discussion, the authors suggest a higher infection rate with resistant strains in summer. What are the arguments for this suggestion (as infection rates are often higher in winter.

Reviewer 2 Report

In their manuscript, Yasmin Saba and Zahra Rabaab assess the antibiotic susceptibility profiles and genetic background  of Meropeneme resistant E.coli isolates from sewage water samples. The samples were obtained at different sites within the city of Islamabad and cover 4 different time points between October 2016 and September 2017. Their results clearly show large variability in the prevalence of resistant E. coli throughout the sampling period. The 124 meropeneme resistant isolates were further caracterised for resistance against additional antibiotic classes, the phylogenetic group these isolates belong to and the presence of several resitance marker genes.

While this work is novel and interesting, some claims from the introduction and title of the manuscript are more ambitious than the data and study design would justify:

  • saisonal variation should, in my oppinion, not be concluded from a single year of observation and I would highly recommend to avoid such a statement in the title
  • while the sampling points were carefully distributed within Islamabad, the actual distance between all sampling points seems not more than approx. 10-12 km limiting the geographic spread and thus the potential differences in abiotic factors contribution to spread of resistant bacteria which ultimately limits the generalisability of the results
  • generally more details on the sampling sites (e.g. water temerature) should be provided
  • measurements of degradation products or residues of antibiotics and their concentration at the sample sites would have greately improved the impact of this study
  • the statement "E. coli being a good AMR indicator due to its extensive presence in wastewater excreting from human and animal gut, can provide an understanding of AMR status in the environment. Thus, by looking into anti-biotic resistant E. coli in the environment we can get insight into community carriage of antibiotics resistant bacteria and the potential hazards for the community." from the introduction needs proof/experimental evidence

In addition to these general points that should critically be revised, there are a few specific points that should be adjusted:

  • the increase in meropeneme resistant E.coli isolated during summer season might be a result of a increased microbial load during this time and should be related to the number of total E.coli cells or even microbial cfu's determined on non-selective plates
  • the manuscript would benefit a lot from a supplemental table listing all sampling sites (incl. a short description) along with the detected meropeneme resistant E. coli isolates for each time point and perhaps also the results from the resistance screening
  • line 92f: "Only summer season E. coli showed resistance to fosfomycin 4% (n=3)." have these three E. coli been isolated from close sites?
  • Figure 2 and the conclusion from section 2.2 (higher diversity of phylogenetic groups during summer) is misleading as way more isolates were obtained during this season and prevalences of e.g. 2% may not be detected in a saison containing ~ 10-20 samples only
  • Figure 5: for visual improvement of the manuscript I suggest to present these data in a venn diagramm (e.g. for area-proportional venn diagrams https://www.deepvenn.com/)
  • line 135: please check that "showed 27% of the sites positive for meropenem resistant E. coli" this is correct. In the results section 27% of all samples (not sites!) contained meropenem resistant E. coli. This might be by coincidence but could also be an error.
  • the comparison to data from studies carried out in India in the discussion section should be complemented by a brief description of the usage of different antibiotics in both countries
